# Genetic and morphometric variability between populations of *Betula ×oycoviensis* from Poland and Czechia: A revised view of the taxonomic treatment of the Ojców birch

**Rostislav Linda** *, Ivan Kuneš, Martin Baláš

Department of Silviculture, Faculty of Forestry and Wood Sciences, Czech University of Life Sciences Prague, Suchdol, Czechia

* lindar@fld.czu.cz

**Data Availability Statement:** All relevant data are within the manuscript and its Supporting information files.

## Abstract

Birches are generally known for their high genetic and morphological variability, which has resulted in the description of many species. Ojców birch was described in 1809 by Willibald Suibert Joseph Gottlieb Besser in Poland. Since then, several studies assessing its taxonomy were conducted. Today, various authors present Ojców birch at different taxonomic ranks. In Czechia, the Ojców birch is classified a critically endangered taxon and confirmed at one locality consisting of several tens of individuals. However, before a strategy for its conservation can be applied, we consider it necessary to assess the taxonomic position of the endangered Czech population and to evaluate its relationship to the original Polish population. This study aimed to evaluate the morphometric and genetic variability between populations of *B. ×oycoviensis* in Poland and the Czechia and their relationship to regional populations of *B. pendula*, one of the putative parental species of the Ojców birch. Altogether, 106 individuals were sampled, including the holotype of *B. szaferi*, the second putative parental species of *B. ×oycoviensis*, received from the herbarium of W. Szafer, which is deposited at the Institute of Botany in Kraków. Morphological analyses identified differences in leaves between *B. ×oycoviensis* and *B. pendula*. However, no significant differences were found in genome size between selected taxa/working units except for *B. pendula* sampled in Czechia. The identified difference of the Czech population of *B. pendula* is probably caused by population variability. Genetic variability between all the taxa under comparison, regardless of their origin, was also very low; only the benchmark taxa (*B. nana* and *B. humilis*) clearly differed from all samples analyzed. The results indicate minute morphological and negligible genetic variability between the Czech and Polish populations of *B. ×oycoviensis*. In light of our results, the classification of *B. ×oycoviensis as B. pendula* var. *oycoviensis* seems more accurate than all hitherto presented alternatives (e.g. *B. ×oycoviensis* as a separate species).

**Funding:** The research activities were supported by Technology Agency of the Czech Republic: grant project No. TACR TH03030339 (Ivan Kuneš) and by Internal Grant Agency of Faculty of Forestry and Wood Sciences No. A_19_24 (Martin Baláš). The funders had no role in study design, data collection and analysis, decision to publish, or preparation of the manuscript.

**Competing interests:** The authors have declared that no competing interests exist.

## Introduction

The genus *Betula* is generally known for its extensive genetic and morphological variability among taxa [1, 2]. The main reasons for such variation are frequent hybridization and subsequent introgression as well as polyploidization [3–5]. As a consequence, ca 64 birch species are currently distinguished worldwide [2]. However, the taxonomic treatment of some birches is ambiguous, and various authors present different numbers of species, usually ranging between 30 and 60 [6, 7]. Several minute birch taxa with questionable taxonomic position were also described in Central Europe in the 19th and 20th centuries, such as *B. carpatica*, *B. obscura*, *B. atrata*, and *B. petraea* [8–11]. One intricate taxon of the genus *Betula* is the Ojców birch (*Betula ×oycoviensis* Besser), which was first described by Besser [12] as *B. oycoviensis*. Ojców birch is a diploid taxon (2n = 28) closely related to silver birch (*B. pendula*). The taxonomic position of the Ojców birch and the character of its relationship to silver birch, however, remains unresolved. A detailed description of the taxon was made available in 1921 [13], and the species *B. oycoviensis* was recognized further in 1928, based on the description of samples taken in Hamernia (Poland) [14]. Since then, several studies focused on determination its origin have been conducted [15–18]. Based on this subsequent research, *B. oycoviensis* was classified as a hybrid between *B. pendula* and '*Betula nova*' [15]. '*Betula nova*' was later described as *Betula szaferi* Jentys-Szaferowa ex Staszkiewicz [19]. Throughout the following text, we refer to Oyców birch using the established scientific name *B. ×oycoviensis* (as did, for example, Rutkowski [20]), although some authors have recently classified this birch at lower taxonomic ranks, for example as *B. pendula* var. *oycoviensis* [21].

*Betula ×oycoviensis* is in many traits similar to *B. pendula*. The most prominent differences can be observed on its leaves, which often grow in groups of 4–6 on brachyblasts and are up to 4 cm long, and in its habitus: Compared to *B. pendula*, *B. ×oycoviensis* generally has the form of lower trees (up to 15 m) or shrubs [17, 20–22] with a curved stem and 'broomy' crown. *Betula ×oycoviensis* has no continuous area of distribution. Besides Poland, some isolated micropopulations of *Betula ×oycoviensis* occur or probably occur in Czechia, Romania, Ukraine, Denmark and Sweden [22–25].

In Poland, several places of occurrence have been reported, for example Dolina Kobylańska (close to Kraków, Lesser Poland Voivodeship), Skielek (Beskid Wyspowy, Lesser Poland Voivodeship) and Czerwona Góra (close to Opatów, Świętokrzyskie Voivodeship) [16, 19, 20]. The number of these places has possibly decreased over the past several decades. The fate of a small group of specimens at Chojnik is not known at present, see the updated version (2013) of a study by Staszkiewicz [26], available at http://archive.is/43WK. In Czechia, there is only one confirmed locality of *B. ×oycoviensis* at the village Volyně u Výsluní in the Ore Mts, North Bohemia [27] and a few proposed unconfirmed localities. Besides the Ore Mts., the Database of the Czech flora and vegetation PLADIAS (https://pladias.cz/en/ to date of 2020/04/07) suggests other possible occurrences of *B. ×oycoviensis* elsewhere in the country—in the Křivoklátsko region (Central Bohemia) and the Třeboňsko area (South Bohemia), see Fig 1.

Some studies assessing the origin of *B. ×oycoviensis* have already been published (e.g. the already mentioned studies by Jentys-Szaferowa [28] or Staszkiewicz [19]). However, detailed studies comparing the genetic variation of *B. ×oycoviensis* and *B. pendula* populations in Central Europe are missing.

This study aims to assess the morphological and genetic variability between Czech and Polish populations of *B. ×oycoviensis* and to compare these populations with local populations of *B. pendula*. The Polish populations of *B. ×oycoviensis* were selected for our comparison because the original locations from which *B. ×oycoviensis* was described are in Poland. Moreover, *B. ×oycoviensis* is often reported to be of hybrid origin [15]. One of its proposed parental

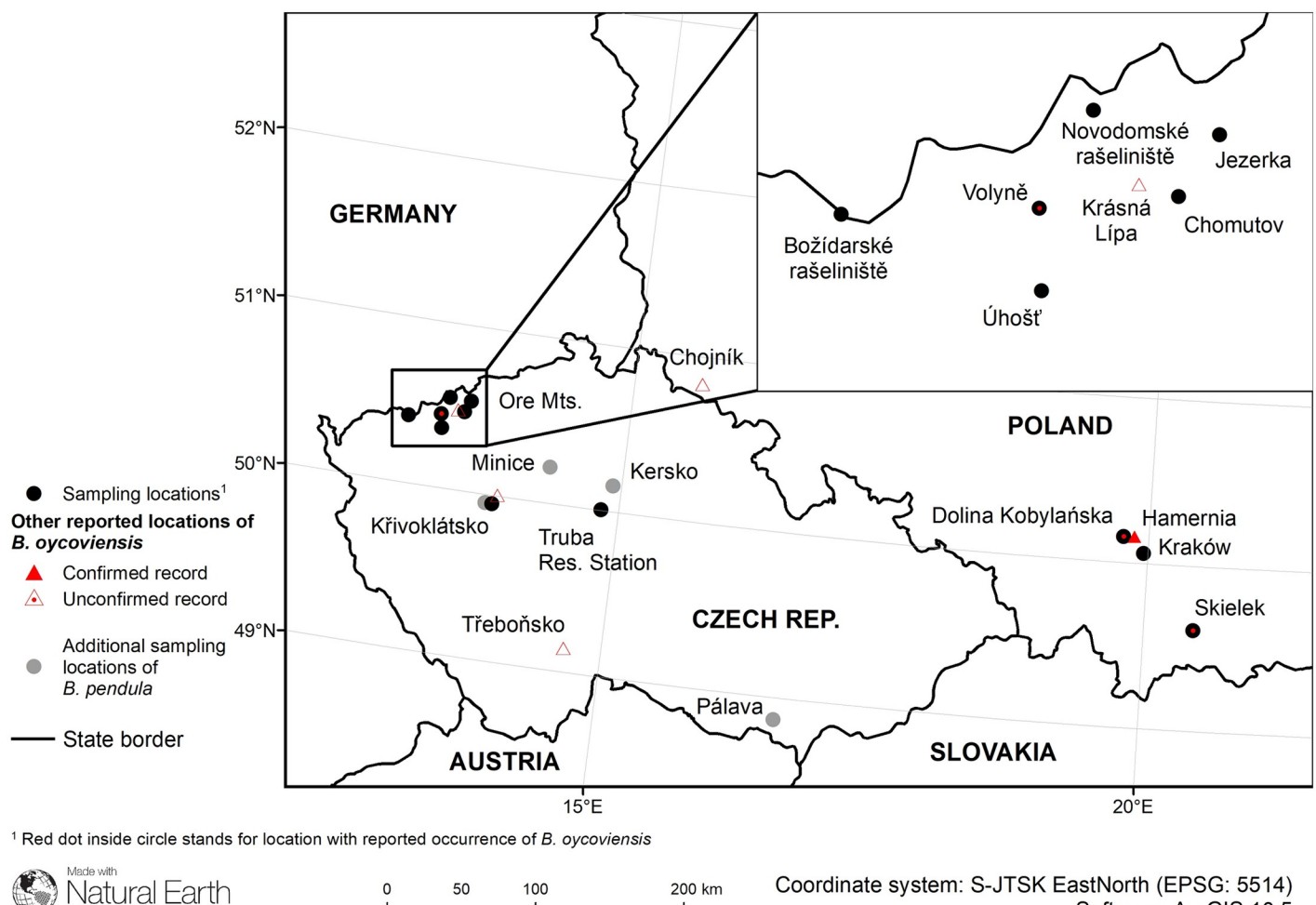

**Fig 1. Distribution of sampling locations along with other reported locations of *B.* ×oycoviensis.** Sample locations for additional cytometric analyses (see the Discussion) are depicted using grey circles. Locations with the acknowledged (generally accepted) occurrence of *B.* ×oycoviensis are marked with filled red triangles (or red dots inside of black circles, if these locations were sampled within our study). Empty red triangles depict locations of proposed (unverified) occurrence of *B.* ×oycoviensis according to Database of the Czech flora and vegetation PLADIAS ([https://pladias.cz/en/](https://pladias.cz/en/) to date of 2020/04/07) and study by Staszkiewicz [18].

species, *B. szaferi*, was available to us (original acknowledged specimen—holotype) only as a herbarium item from the collections in Kraków (Poland), see Material and methods.

Besides the taxonomic perspective, such studies are also desirable from a practical standpoint. Management steps should be taken towards the effective conservation of the Czech Ojców birch population because it is growing old and its natural regeneration is poor. However, these steps should reflect the taxonomical position of the Ojców birch, so it is important to determine whether its Czech population is taxonomically identical to that in Poland. If it is, Polish individuals could theoretically be used to strengthen the Czech population.

For assessing the variability between *B.* ×oycoviensis and closely related *B. pendula* in Poland and Czechia, we employed leaf morphometry, genome size analysis and microsatellite analysis. Previously, leaf morphometry was employed for the study of relationships between birch taxa, for example by Gardiner et al. [29], Atkinson [30] or Gill and Davy [31], whose studies served as a basis for the selection of parameters used in this study. The presented microsatellite analysis was recently used for the determination of taxonomic relationships of

the Carpathian birch [32] and was selected based on studies by Kulju et al. [33] and Tsuda et al. [34, 35].

## Material and methods

### Plant material

Individuals of *B. ×oycoviensis* for our study were sampled between 2017 and 2019 at Dolina Kobylańska and Skielek in Poland and at Volyně u Výsluní (in the Ore Mts) in Czechia. The specimens of *B. ×oycoviensis* were complemented by samples taken in the Botanical Garden of the Jagiellonian University (BGJU) in Kraków (Poland) and at Chomutov ZOO (Czechia; the sampled trees were transplanted from Volyně in the past). One birch showing several traits of *B. ×oycoviensis* was found and sampled in the Křivoklátsko region (Czechia). Reference samples of *B. pendula* originate from the localities Božídarské rašeliniště, Jezerka, Novodomské rašeliniště and Úhošť in the Ore Mts. (Czechia), and Skielek (Poland). The Czech samples of *B. pendula*, together with a sample of *B. nana* described below, were collected in 2012 and used in our previous study [32]. As benchmark species for the evaluation, we included two additional birch taxa (*B. nana*, also collected in 2012 at Božídarské rašeliniště in the Ore Mts., Czechia, ca. 24 km to the west of the town of Volyně, and *B. humilis*, which was provided by the BGJU). The exact location (using Leica GPS 1200 at Volyně and Garmin GPSMAP 62S in other cases) and assignment of each sampled individual was recorded in the field. Specific individuals showing some traits of *B. szaferi* were cultivated at the Research Station Truba near the town of Kostelec nad Černými lesy in Czechia ('cult. *B. szaferi*' for further reference). The ancestors of the 'cultivated *B. szaferi*' specimens from Truba were individuals of *B. ×oycoviensis* growing at Volyně u Výsluní, whose seeds were collected in 2017. One dry leaf of the holotype (KRAM 303846) of *B. szaferi* [19] was provided by the herbarium of the W. Szafer Institute of Botany of the Polish Academy of Science (PAS) in Kraków (its photo is included as S1 Fig).

The determination of the taxa was done in the field as follows: As *B. ×oycoviensis* we determined individuals that exhibited at least 80% of the traits described in the relevant literature [16, 19], and the rest were determined as *B. pendula*. All other transitional individuals, which combined traits of the two taxa, were determined as *B. pendula*/*B. ×oycoviensis* (a 'working unit' *sensu* Kuneš *et al.* [32]). Besides the benchmark species *B. nana* and *B. humilis*, we also collected sample material from *B. atrata* and *B. obscura*, which are visually distinct dark-barked birches occurring at the sites of sampling (conducted in 2019), although the taxonomic significance of these dark-barked birches remains questioned. We determined these taxa on the basis of morphological traits described by Domin [10], Hejtmánek [36] and Franiel [37]. Individuals of cult. *B. szaferi* were classified according to traits referred by Staszkiewicz [19]. However, in contrast to the holotype specimen of *B. szaferi* from the herbarium in Kraków, the individuals of 'cult. *B. szaferi*' grown at Truba should be viewed more as a working unit than as an acknowledged species. The determination of the remaining taxa (*B. nana* and *B. humilis*) is straightforward. From most of the sampled tree individuals, we took three short branches (two for morphometric analysis and one for flow cytometry and molecular analyses).

In total, we sampled twelve taxa and working units from twelve localities (Table 1, Fig 1). We provided a simple table with basic description of all involved taxa in this study as (S1 Table). To simplify the text, we use the term 'group' to refer to each taxon or working unit with the country of origin. The limited number of specimens of *B. ×oycoviensis* in the study is given by the fact that the wild Czech and Polish populations are small. Moreover, the individuals with sufficiently manifested traits of Ojców birch are scattered in stands of *B. pendula* and other species, and their proportion is low. Even in BGJU in Kraków, only a few trees retained

**Table 1. Numbers of individuals sampled at individual localities in Czechia and Poland.**

| State | Region | Locality | GPS Coordinates (WGS84) | | Group (taxon/working unit) | Number of sampled individuals |
|---|---|---|---|---|---|---|
| | | | *Latitude* | *Longitude* | | |
| **Czechia** | Ore Mts. and surroundings | Božídarské rašeliniště | 50.408 | 12.908 | *B. nana* | 1 |
| | | | | | *B. pendula* (CZ) | 1 |
| | | Chomutov[1] | 50.478 | 13.431 | *B. ×oycoviensis* (CZ) | 2 |
| | | Jezerka | 50.546 | 13.481 | *B. pendula* (CZ) | 9 |
| | | Novodvorské rašeliniště | 50.551 | 13.277 | *B. pendula* (CZ) | 2 |
| | | Úhošť | 50.363 | 13.239 | *B. pendula* (CZ) | 8 |
| | | Volyně[2] | 50.445 | 13.216 | *B. ×oycoviensis/B. pendula* (CZ) | 4 |
| | | | | | *B. ×oycoviensis* (CZ) | 33 |
| | Křivoklátsko | Bušohrad | 49.951 | 13.813 | *B. ×oycoviensis/B. pendula* (CZ) | 1 |
| | Truba Research Station | Truba | 50.006 | 14.836 | 'cult. *B. szaferi*'[3] | 3 |
| **Poland** | | Kobylany[2] | 50.155 | 19.760 | *B. atrata* | 1 |
| | | | | | *B. obscura* | 1 |
| | | | | | *B. ×oycoviensis/B. pendula* (PL) | 1 |
| | | | | | *B. ×oycoviensis* (PL) | 16 |
| | | | | | *B. pendula* (PL) | 5 |
| | | Kraków (botanical garden[4]) | 50.062 | 19.959 | *B. humilis* | 1 |
| | | | | | *B. ×oycoviensis/B. pendula* (PL) | 2 |
| | | | | | *B. ×oycoviensis* (PL) | 2 |
| | | Skielek[2] | 49.613 | 20.464 | *B. ×oycoviensis* (PL) | 7 |
| | | | | | *B. pendula* (PL) | 5 |
| | | Kraków (herbarium[5]) | 50.066 | 19.995 | *B. szaferi*[6] (herbarium) | 1 |
| **TOTAL** | | | | | | **106** |

[1]Specimens of the Czech origin transplanted from Volyně;

[2]locality considered as a place of the natural occurrence of *B. ×oycoviensis;*

[3]specimens with some traits of *B. szaferi* cultivated from seeds of the Ojców birches in Volyně;

[4]The Botanical Garden of the Jagiellonian University;

[5]Herbarium of W. Szafer Institute of Botany, Polish Academy of Sciences;

[6]Holotype (KRAM 303846) of *B. szaferi*.

traits of the Ojców birch to such an extent that it allowed us to classify them as unequivocally identifiable *B. ×oycoviensis*.

The sampling of *B. humilis* was permitted by document no. DZP-WG.6400.6.2020.EP.2 (Generalny dyrektor ochrony Środowiska, Warsaw) and sampling of *B. ×oycoviensis* was permitted by documents no. OP-I.6400.15.2018.KW (Regionalny dyrektor ochrony Środowiska, Krakow) and no. DZP-WG.6400.23.2018.ep (Generalny dyrektor ochrony Środowiska, Warsaw).

## Morphological analyses

Whenever possible, we collected two branches from each sampled individual for analysis of leaf morphology using telescopic shears. From each branch, we randomly selected two leaves for taking measurements (in total, we assessed four leaves per each analyzed individual). We measured following traits on each leaf: blade length [mm], blade width [mm], blade fitting angle [˚], blade tip angle [˚], leaf serration angle [˚], petiole length [mm], distance of the widest part of the blade from the blade base [mm], number of leaf veins [–], distance between the 3rd

and 4[th] vein [mm], number of leaf teeth between the 3[rd] and 4[th] vein [–], blade width in the upper 1/4 [mm], distance from the leaf base to the 1[st] tooth [mm], basal angle [°], 1[st] vein angle [°], 4[th] vein angle [°] and distance from the 4[th] vein to the tip [mm]. We calculated the final values for each individual and each parameter were calculated as averages of four measurements. In the cases of samples of 'cultivated *B. szaferi*' from the Truba Research Station we obtained and averaged only two measurements because it was not possible to take two branches from these samples (the individuals were too young), and in the case of *B. szaferi* from the herbarium of PAS we obtained only two measurements because we used a scanned image of the herbarium specimen. We analyzed this specimen by taking measurements of its image using the software ImageJ 1.52 [38]. We did not measure the morphological features of the benchmark taxa (*B. nana* and *B. humilis*), because it was not possible to measure all parameters on the leaf (4[th] veins were not present on their leaves). Except for *B. szaferi* (Herbarium), we measured all specimens using a simple ruler and protractor. We did not include some individuals in the morphological analyses, because it was not possible to take enough material under field conditions; for example, the crown was sometimes too high to allow the sampling of sufficiently developed, insolated leaves for morphometric measurements (e.g. *B. atrata* and *B. obscura* in our study). Further methodological information on the measuring of morphological parameters of birch leaves are provided by previous studies [39, 40]. In total, we selected 87 individuals for morphological analyses.

## Genome size analysis

The genome size of sample plants was determined by propidium iodide flow cytometry [41] with *Solanum pseudocapsicum* L. (2C = 2.61 pg) as the internal standard. Leaf tissue from two petioles of each sample was chopped together with about 1.5 cm$^2$ of *S. pseudocapsicum* leaf tissue in 0.5 ml of Otto I buffer [42]. The resulting suspension was filtered through a 42-μm nylon mesh and left still for ca 20 minutes at 20°C. After that, the suspension was stained with a solution of the following composition: 1 ml of Otto II buffer [42], β-mercaptoethanol (2μl/ml), propidium iodide and RNase IIA.

Flow cytometry was performed using a Partec CyFlow flow cytometer (Partec, Germany) equipped with a green solid-state laser (Cobolt Samba, 532 nm, 100 mW). Holoploid genome size and 1Cx-values (i.e. holoploid genome size divided by the number of chromosome sets [43]) was computed from raw cytometric data using FloMax software and evaluated statistically.

## Molecular analyses (SSR genotyping)

Before DNA extraction, individual samples (leaves), stored at −80°C, were frozen in liquid nitrogen and ground by an oscillation mill Retsch MM400 (Retsch GmbH, Haan, Germany). DNA was then extracted from the powdered samples using the QIAGEN DNEasy Plant Mini Kit (QUIAGEN, Hilden, Germany) following the standard protocol. The concentration and quality of DNA was checked prior to PCR using a NanoDrop 2000 (ThermoFisher Scientific, Waltham, Massachusetts, USA) spectrophotometer. For PCR, each DNA sample was diluted to a concentration of 10 ng/μl.

The genetic diversity of the taxa under study was assessed by microsatellite analysis of nuclear DNA. Microsatellite markers were selected based on publications by Tsuda *et al.* [34, 35] and Kulju *et al.* [33]. In total, 50 markers were tested, from which 12 polymorphic loci were selected (see S2 Table) and optimized for two multiplex groups, identically as in a previous study by Kuneš *et al.* [32].

The PCR reactions were run in a total volume of 20 μl: 15 ng of DNA, primers (0.25 μM of each primer), 200 μM of dNTP, 2.5 mM MgCl$_2$ and 1 μM buffer PCR multiplex mix with polymerase.

The PCR program according to Kulju *et al.* [33] consisted of an initial 5 minutes of denaturation (95˚C), which activated the hotStarTaq polymerase, followed by 30 cycles of denaturation (95˚C, 60 s), annealing (57˚C, 75 s) and elongation (72˚C, 150 s). The final extension step took 10 minutes (72˚C).

In the PCR program following Tsuda *et al.* [34, 35], 30 cumulative cycles were performed: denaturation (95˚C, 30 s), annealing (55˚C, 30 s) and elongation (72˚C, 45 s) with a final extension step (7 minutes, 72˚C).

The volume of 1 μl of PCR product was added to 14 μl of a solution of formamide with GeneTrace500 DNA ladder (Carolina Biosystems, Prague, Czechia), prepared according to the manufacturer's protocol. The solution was denatured (5 minutes, 95˚C) and quickly cooled on ice.

For determining the length of the amplicons containing the microsatellites, the genetic sequencer Genetic Analyser 3500 (Applied Biosystems, Foster City, California, USA) was used. Raw sequence data were analyzed by GeneMapper 4.1 software (Applied Biosystems, Foster City, California, USA).

## Statistical analyses and computations

**Morphometric analysis.** To compare foliar features between groups (taxa/working units), we performed multivariate analysis of variance (MANOVA) and canonical discriminant analysis (CDA). The results of the CDA are presented as scatterplots. We evaluated the accuracy of CDA discrimination based on ratios of successfully classified individuals for each working group. We performed all computations in R software [44] and produced scatterplots using the R package 'ggplot2' [45] and the MorphoTools function set for R [46].

We tested for differences in individual parameters between unequivocally identifiable *B. pendula* and *B. ×oycoviensis* individuals across all populations taken together using the t-test (when all assumptions were met) or the Wilcoxon rank-sum test (when they were not). We evaluated the normality of our data was using the Shapiro-Wilk normality test. When the assumption of normality was met but the datasets differed significantly in variances (tested by the F-test of equality of variances), we used Welch test instead. We performed all statistical computations in R [44], with the significance level set to α = 0.05.

**Genome size analysis.** We did not evaluate the benchmark species (*B. nana*, *B. humilis*) and dark-barked birches (*B. atrata* and *B. obscura*) in this analysis, as only a few samples of these taxa were available. The *Betula szaferi* sample obtained from the Kraków herbarium was analyzed without success, as it was impossible to perform flow cytometry on this old dry sample. Flow cytometry was also not successful for five samples of *B. pendula* from Czechia, one sample of *B. ×oycoviensis* from Poland, two samples of *B. ×oycoviensis* from Czechia and one sample of 'cultivated *B. szaferi*'. Therefore, flow cytometric analysis was accomplished in 92 samples.

Differences in 1Cx values between working groups were tested for by a Kruskal-Wallis test with multiple comparisons [47], as the assumptions of ANOVA were not met. A box-plot presenting statistically significant differences was produced in R software [44] using the package 'ggplot2' [45]. All statistical tests were performed using the significance level of α = 0.05.

**Molecular analyses.** Genotype data were checked for errors and invalid records. Basic parameters (total allele count, number of used loci and total number of individuals) were evaluated together with basic *F*-statistics ($F_{ST}$, $F_{IS}$, $F_{IT}$) [48] between selected taxa (working

groups). For each of the selected taxa (working groups), the numbers of private alleles (those which did not occur in any other group at the respective loci) across all loci were also evaluated.

To visualize relationships between individuals in the study, discriminant analysis of principal components (DAPC) [49] was performed. The structure of the dataset was also evaluated by clustering analysis in STRUCTURE 2.3.4 [50–52] using the following settings: K– 1 to 15 (number of groups + 3), number of runs for each K– 30, burn-in– 100,000 repeats, MCMC repeats– 100,000 repeats. An admixture model with correlated allele frequencies was used. The remaining parameters were left at their default settings. The actual number of populations in the dataset was determined by the method presented by Evanno [53] using Structure Harvester [54]. Cluster matching and permutation was performed in CLUMPP 1.1.2, utilizing the Greedy algorithm with 1,000 repeats [55]. STRUCTURE plots were produced using DISTRUCT [56]. Testing for differentiation between selected taxa (working groups) was done using Goudet's G-statistic Monte Carlo test [57].

All computations except for those performed in STRUCTURE, CLUMPP and DISTRUCT software were done in R [44], using the packages 'adegenet' [58, 59], 'poppr' [60, 61] and 'hierfstat' [62]. R plots were made using the package 'ggplot2' [45].

## Results

### Morphometric analysis

As is apparent from the CDA scatterplot (Fig 2), there was relatively low variability in morphological traits of leaves among most of the individuals except for the sample of *B. szaferi* (taken from the herbarium), which is clearly separated on the CDA scatterplot. When the original sample of *B. szaferi* was excluded from the analysis, samples of 'cultivated *B. szaferi*' appeared outside the point cloud. After the removal of *B. szaferi*, also the distinction between *B. ×oycoviensis* and *B. pendula*, and to some extent between the Czech and Polish populations of *B. pendula* and *B. ×oycoviensis*, became more visible. Significant differences between *B. pendula* and *B. ×oycoviensis* were identified in several leaf parameters by statistical analyses (Table 2).

The testing for differences in leaf parameters between selected working groups yielded significant results (MANOVA, df = 7, Pillai's trace = 2.902, p < 0.001). The accuracy of CDA discrimination was 100% except for *B. ×oycoviensis* from the Czechia (74.2%), *B. ×oycoviensis* from Poland (81.3%) and *B. pendula* from Poland (70.0%).

Trees classified as *B. pendula* and *B. ×oycoviensis* (those which showed at least 80% of traits described in the relevant literature, see the Material and methods) differed significantly (p < 0.05) from each other in 13 out of the 16 morphometric parameters analyzed (see Table 2).

### Genome size analysis

Individuals of *B. pendula* from Czechia (the Ore Mts. and surroundings) possessed greater 1Cx values (median: 0.4413) compared to the other groups analyzed, including Polish *B. pendula* (median for the other groups together: 0.4280). The Kruskal–Wallis test revealed significant differences (chi-squared = 32.313, df = 6, p < 0.001) in 1Cx values. The results of multiple comparisons are depicted in Fig 3.

### Molecular analyses

Altogether, 116 alleles at twelve loci were identified in a total of 106 individuals. *F*-statistics indicated very low overall variability ($F_{IT} = 0.1088$), 23% of which accounted for among-population variability ($F_{ST} = 0.0254$, $F_{IS} = 0.0834$). After the removal of the two benchmark taxa (*B. nana* and *B. humilis*) from the dataset (see below for more details), $F_{ST}$ accounted for only 12%

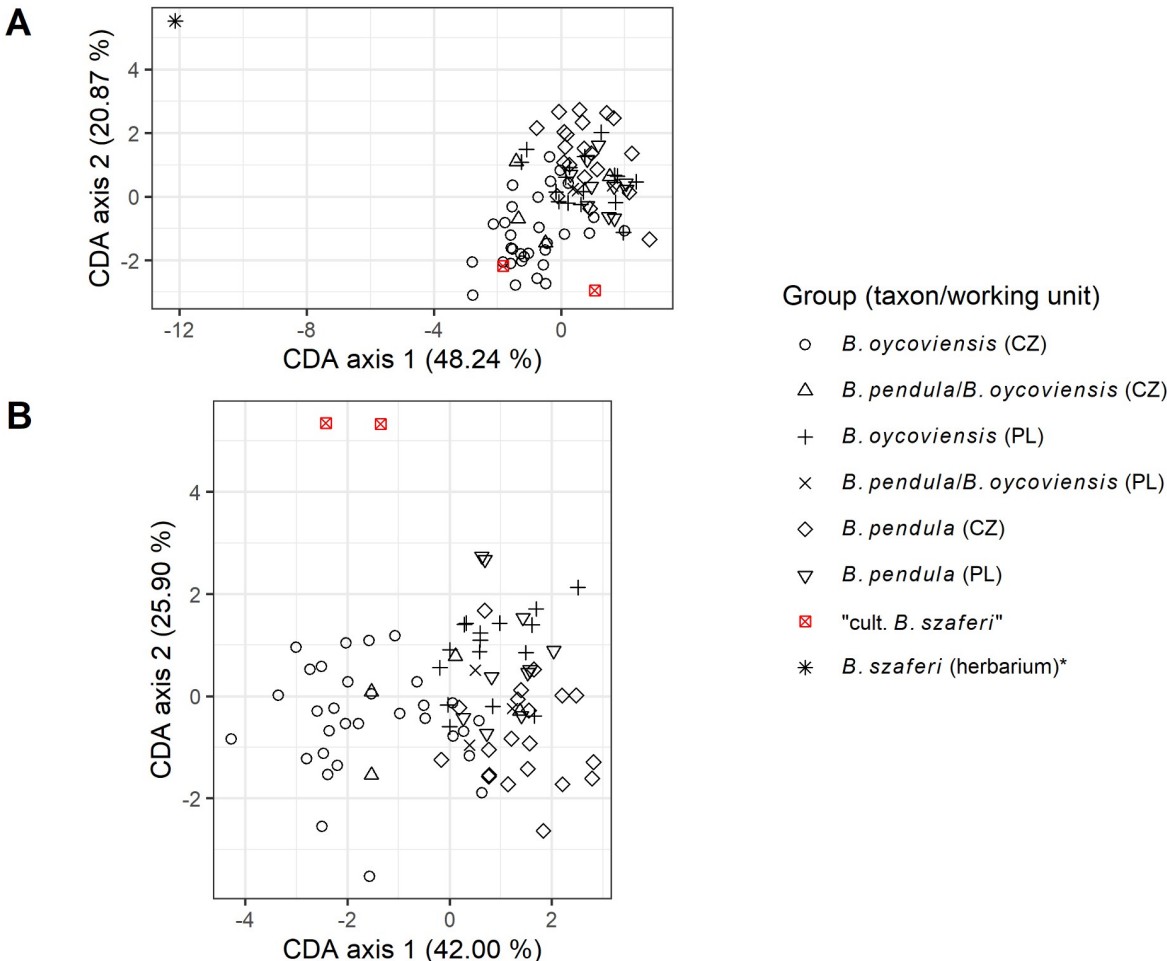

**Fig 2. Results of canonical discriminant analysis (CDA) of morphological traits of leaves presented as a scatterplot of selected birch individuals.** Plot A depicts the results of CDA analysis of all samples included and plot B depicts the results of CDA analysis without the holotype of *B. szaferi*, which appeared as an outlier in the first CDA analysis. The percentages in axis titles stand for percentages of explained variation by the respective axis.

of the variability, suggesting that all the other groups form quite a homogeneous complex ($F_{ST}$ = 0.0105, $F_{IS}$ = 0.0799).

Private alleles were found in seven groups out of the twelve groups examined. The greatest numbers of private alleles per individual were found in the cases of *B. humilis* and *B. nana* (benchmark taxa). Among the other groups, the greatest number (proportion) of private alleles per individual was found for *B. pendula* from Czechia (Table 3).

The overall DAPC analysis (with all samples included in the analysis) clearly separated the benchmark taxa (*B. nana* and *B. humilis*) whereas all other groups formed a compact cloud with relatively low variation (Fig 4).

The homogeneity between all groups (working units) except for *B. nana* and *B. humilis* was also supported by analyses in STRUCTURE and CLUMPP software. The number of clusters in the whole dataset was set to 4, based on an analysis according to Evanno *et al.* [53] in Structure Harvester [54]; for more information, see S2 Table. The benchmark taxa were included in a separate group (denoted by orange color in Fig 6) whereas all other groups formed quite a homogeneous cluster.

**Table 2. Results of testing for differences between *B. pendula* and *B. ×oycoviensis* in individual foliar parameters.**

| Parameter | Mean (*B. oyc.*) | Mean (*B. pen.*) | Test | Test stat. value | p-value | Sig. |
|---|---|---|---|---|---|---|
| Blade length [mm] | 33.9 | 43.1 | N | 239.5 | < 0.001 | *** |
| Blade width [mm] | 25.6 | 33.9 | N | 235.0 | < 0.001 | *** |
| Blade fitting angle [°] | 292.6 | 299.6 | N | 553.5 | 0.115 | ns. |
| Blade tip angle [°] | 47.5 | 40.1 | P* | 3.485 | <0.001 | *** |
| Leaf serration angle [°] | 74.6 | 63.2 | P* | 3.751 | < 0.001 | *** |
| Petiole length [mm] | 15.2 | 15.8 | P | 0.823 | 0.413 | ns. |
| Distance of widest part of blade from blade base [mm] | 11.6 | 13.7 | N | 440.0 | 0.006 | ** |
| Number of leaf veins [–] | 6.0 | 7.8 | N | 204.0 | <0.001 | *** |
| Distance between 3rd and 4th vein [mm] | 4.8 | 6.0 | N | 294.0 | <0.001 | *** |
| Number of leaf teeth between 3rd and 4th vein [–] | 1.6 | 2.7 | N | 289.0 | <0.001 | *** |
| Blade width in the upper 1/4 [mm] | 10.2 | 12.0 | N | 403.0 | 0.002 | ** |
| Distance from the leaf base to the 1st tooth [mm] | 11.7 | 14.3 | N | 291.0 | < 0.001 | *** |
| Basal angle [°] | 245.3 | 239.2 | N | 889.5 | 0.055 | ns. |
| 1st vein angle | 56.4 | 65.1 | P* | 3.532 | <0.001 | *** |
| 4th vein angle | 30.3 | 32.5 | P | 2.109 | 0.038 | * |
| Distance from the 4th vein to the tip [mm] | 19.1 | 31.0 | N | 197.5 | <0.001 | *** |

Notes: N in the column 'Test' stands for the non-parametric Wilcoxon rank-sum test, P stands for the parametric t-test. P* stands for the Welch t-test. Significance level symbols: ns.–not significant,

*–$p < 0.05$,

**–$p < 0.01$,

***–$p < 0.001$.

Only clearly distinguishable individuals were analyzed. Samples from Czechia and Poland were analyzed together. In total, 47 samples of *B. ×oycoviensis* and 30 samples of *B. pendula* were analyzed.

## Discussion

The Ojców birch (*B. ×oycoviensis* Besser) is one of the birch taxa whose taxonomic treatment remains unresolved. It was first described by Besser [12], and studies carried out in the 20th century [15–17] suggested that *B. ×oycoviensis* is a hybrid species between *B. pendula* and '*B. nova*' [15]. The latter putative parent was later validly described as *Betula szaferi* by Jentys-Szaferowa ex Staszkiewicz. By giving the epithet '*szaferi*' to '*B. nova*', Jentys-Szaferowa wished to commemorate her husband, Polish botanist Władysław Szafer [19].

Like Staszkiewicz [19], we found, at sites of reported occurrences of *B. ×oycoviensis* the individuals showing the traits of *B. ×oycoviensis* scattered across stands of typical *B. pendula* or mixtures of typical *B. pendula* and other tree genera (e.g. *Picea*, *Acer* and *Fagus*). Our search for *B. szaferi* in the wild of Czechia and Poland was not successful.

However, we examined several individuals resembling *B. szaferi* that were grown from seeds of specimens classified as *B. ×oycoviensis*, sampled in Czechia (locality Volyně u Výsluní). Similar results were obtained by Jentys-Szaferowa [28], who described three types of progeny obtained by controlled crossings of *B. ×oycoviensis*: type *oycoviensis*, type *pendula* and type '*nova*' (lately '*szaferi*'). Both *B. ×oycoviensis* and *B. szaferi* were originally classified as species. The existence of three types of progeny could imply a 'hybrid' origin of *B. ×oycoviensis* but is definitely not sufficient to prove such a hypothesis. Moreover, no difference between *B. szaferi* and *B. pendula* have been observed using molecular methods (see Figs 4–6). Therefore, *B. ×oycoviensis* could have originated by the crossing of visually somewhat distinct parents belonging to the population of *B. pendula*. This theory would also explain the occurrence of

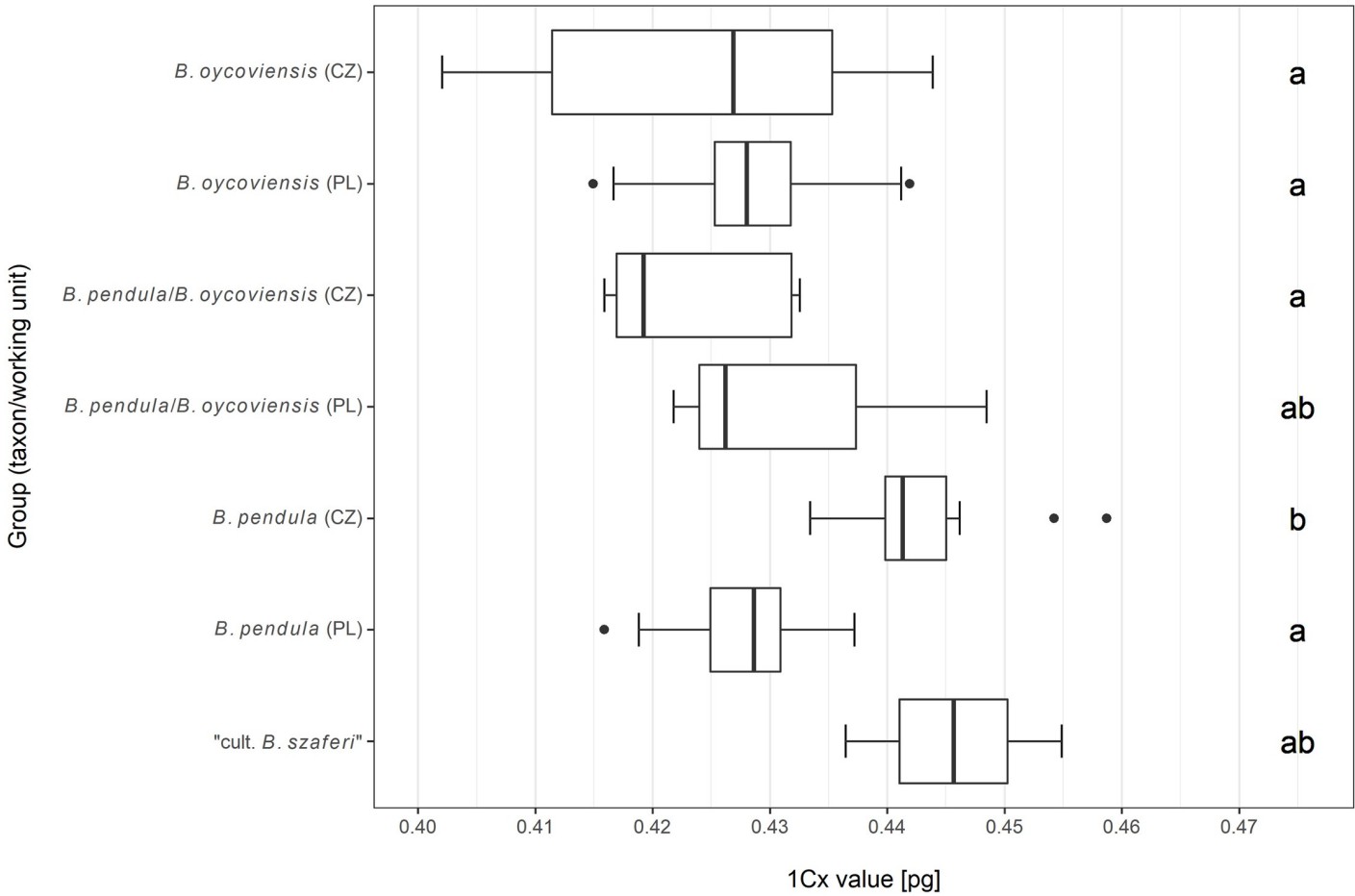

**Fig 3. 1Cx values of samples of selected taxa/working units with results of multiple comparisons.** Groups (taxa / working units in the Poland and Czechia) denoted by different letters exhibited significant differences in 1Cx values at α = 0.05. The box and whiskers plots are presented in standard Tukey's design: Whiskers depict the minimum and maximum excluding outliers, black dots represent outliers (less than the lower quartile—1.5 times the inter-quartile range and more than the upper quartile + 1.5 times the inter-quartile range, respectively).

*B. ×oycoviensis* in countries other than Poland. The question is what traits actually differentiate species from taxa of lower taxonomical ranks, which is rather a conceptual issue [63, 64], and how these traits are controlled and linked.

The testing for differences in leaf morphology via MANOVA yielded significant results. Subsequent CDA analysis revealed distinct separation of the sample of *B. szaferi* taken from

**Table 3. Private alleles of selected groups across all twelve loci.**

| Group (taxon/working unit) | Private alleles | Individuals | Private alleles per individual |
|---|---|---|---|
| *B. ×oycoviensis* (CZ) | 6 | 35 | 0.17 |
| *B. ×oycoviensis* (PL) | 2 | 25 | 0.08 |
| *B. pendula* (CZ) | 10 | 20 | 0.50 |
| *B. pendula* (PL) | 1 | 10 | 0.10 |
| *B. humilis* | 5 | 1 | 5.00 |
| 'cult. *B. szaferi*' | 2 | 3 | 0.67 |
| *B. nana* | 4 | 1 | 4.00 |

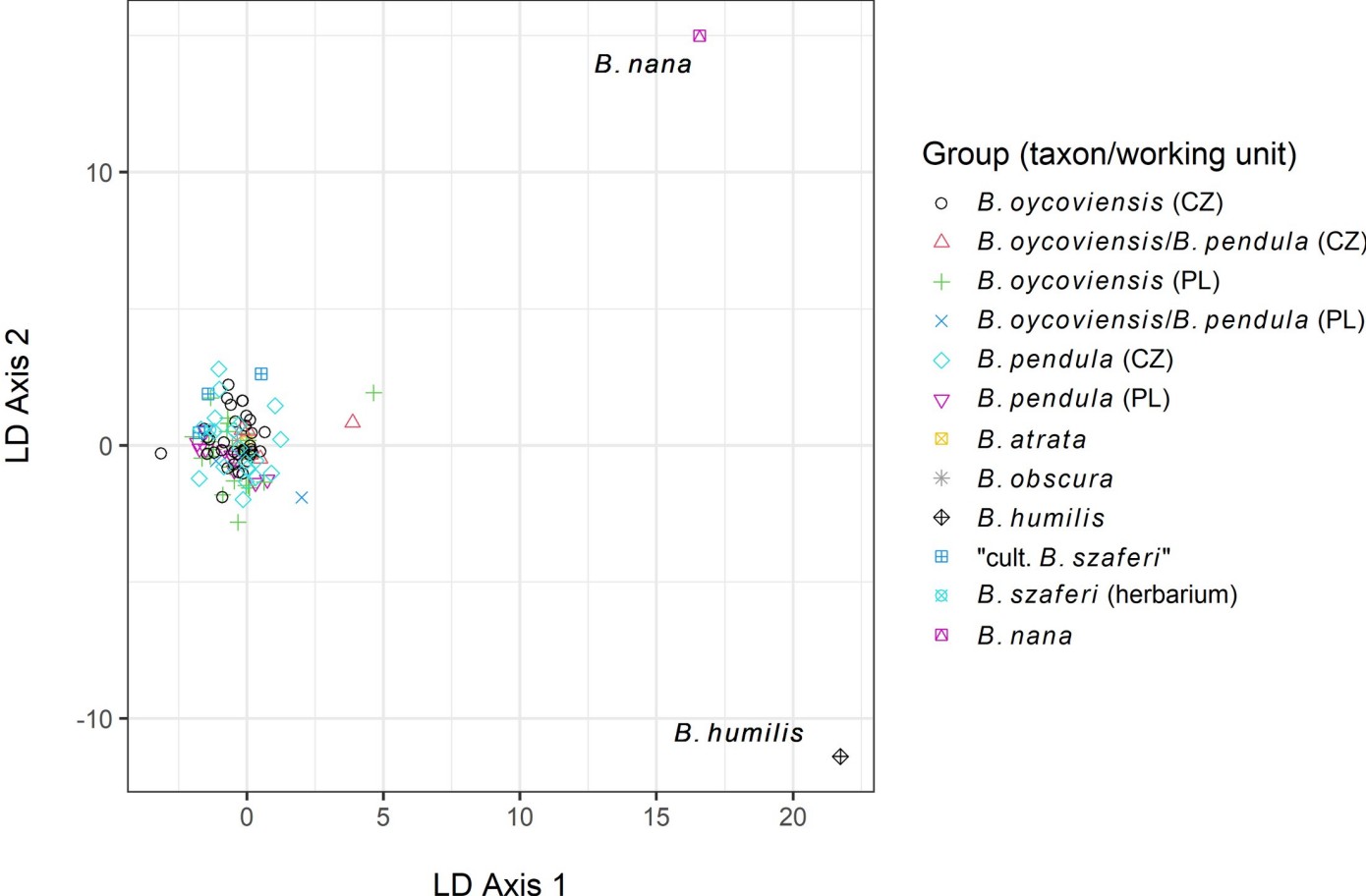

**Fig 4. Scatterplot from DAPC analysis of all individuals. Benchmark taxa, which were the only taxa to be clearly distinguished by the DAPC, are labelled in the plot.** When the two outlier points representing benchmark taxa were removed from the analysis, no clear pattern between the other taxa was detected (Fig 5).

the Krakow herbarium. The separation of this particular sample is not surprising, as the peculiarity of its leaf shape is apparent to the naked eye. When the sample of *B. szaferi* was removed from the CDA analysis, samples of 'cultivated *B. szaferi*' appeared as outliers, which could be expected too, as those samples were selected based on their distinctive leaf morphology. The overall leaf shape of the samples of *B. pendula* and *B. ×oycoviensis* (including samples with mixed traits) showed some minor differences; for example the Czech population of *B. ×oycoviensis* as that of *B. pendula* appeared to be slightly distinctive from its Polish counterparts. This could be a 'random effect' of geographical distance, but it could also be a result of minor phenotypic and genetic differences, possibly reflected in slightly different genome size of the Czech population of *B. pendula* compared to that of other selected working units (see below). This topic, however, surely merits a detailed study, and our data are not sufficient to draw any definite conclusion.

Our testing of individual morphological traits revealed significant differences between clearly identifiable samples of *B. pendula* and *B. ×oycoviensis* in thirteen parameters out of the sixteen tested, namely all parameters tested except for blade fitting angle [°], petiole length [mm] and basal angle [°]. These outcomes of our morphological studies are more or less consistent with the results obtained by Baláš *et al.* [27], who found significant differences in seven

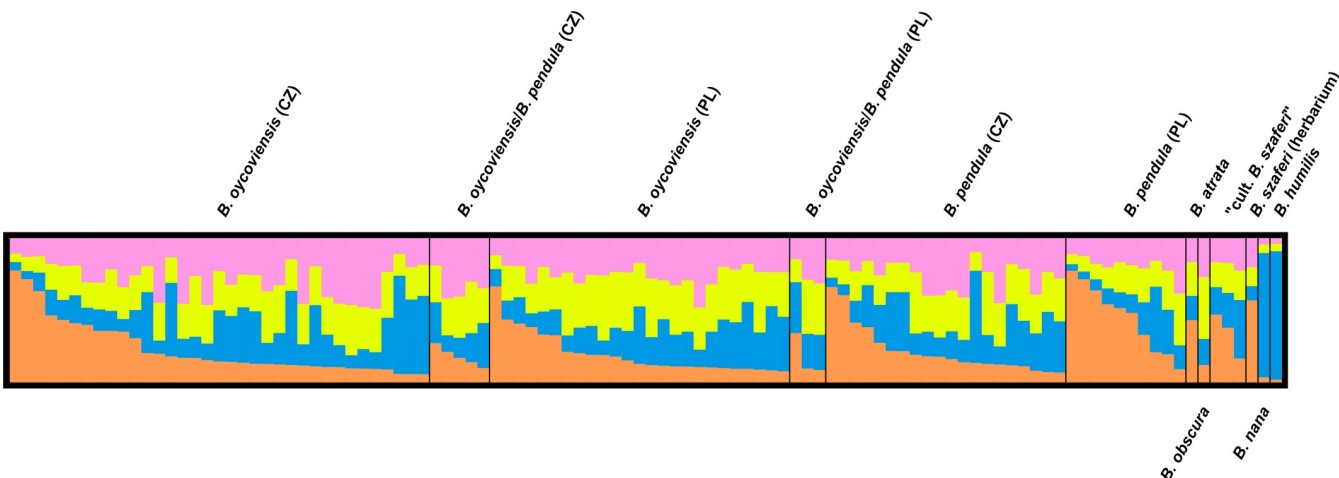

**Fig 5. Scatterplot of individuals excluding benchmark taxa (*B. nana* and *B. humilis*) produced by the DAPC method.** Differences between the groups in the dataset without the benchmark taxa were statistically tested for by G-statistics, which returned non-significant result (100,000 iterations, *p* = 0.68).

**Fig 6. Results of analyses in STRUCTURE and CLUMPP software.** Each bar depicts one individual and its estimated probability of affiliation to a group denoted by its color. The number of groups in the dataset was estimated to be 4 using the method of Evanno *et al.* [53] using STRUCTURE HARVESTER [54]. The plot was created by DISTRUCT software [56].

out of sixteen morphological traits, but it is important to mention that the study by Baláš *et al*. [27] was performed only on samples collected at Volyně u Výsluní, Czechia. Blade length and the number of leaf veins are also mentioned as important for the determination of *B. ×oycoviensis* by the keys to the floras of Poland [20] and Czechia [65], together with the growth of leaves in groups on brachyblasts in *B. ×oycoviensis*. An overview of the whole habitus is needed for the determination of *B. ×oycoviensis* in the field, as the leaf shape differences between the taxa in question do not have to be distinctive enough. Some habitus traits are also described in the keys to the floras mentioned above. Still, many individuals exhibit mixed traits of *B. pendula* and *B. ×oycoviensis*, so it is quite difficult to classify them [66]. That is also the main reason why we distinguished the *B. pendula*/*B. ×oycoviensis* working unit in this study.

Our genome size analysis revealed no significant differences between the defined groups except for *B. pendula* originating from the Czech population (the Ore Mts. and their surroundings), which displayed significantly greater values compared to the other groups. Because *B. pendula* from Czechia does not differ from other groups in the results of our molecular analyses, the reason behind the differences in genome size could reside not only in intraspecific ploidy-related or monoploid-level variation [67] but also in some kind of chromosomal disorder. We, however, attribute these differences to intraspecific variability because our as yet unpublished data indicate relatively extensive variation in genome size (1Cx values, respectively) between populations of *B. pendula* in Czechia (see Fig 7). However, the difference could easily be also a result of some level of methodological inaccuracy. Therefore, a survey focused on genome size across *B. pendula* populations may be desirable to check for the existence of a pattern of genome size with respect to geographic distribution.

Our molecular analyses detected relatively low inter-group variability, which suggests that all of the groups considered are genetically highly homogeneous (including the Polish and Czech *B. ×oycoviensis* and *B. pendula*). The same results concerning *B. ×oycoviensis* were reported in a previous paper by Kuneš *et al.* [32], based, however, only on a small number of *B. ×oycoviensis* samples from Czechia. The only clearly distinguished samples were those of the diploid benchmark species (*B. humilis* and *B. nana*), which were visibly separated from the other samples in the DAPC scatterplot (Fig 4). When these benchmark taxa were removed, only small differences could be observed, for example between samples of 'cultivated *B. szaferi*' and samples of *B. ×oycoviensis* from Poland or between samples of *B. ×oycoviensis* from Czechia and Poland. However, in spite of some gentle difference, the Polish and Czech populations of the Ojców birch most probably belong to the same taxon. The low-level variation is probably caused by some population variability or random effects.

Aside from the benchmark taxa, the greatest mean number (proportion) of private alleles per sample was found in case of *B. pendula* sampled in Czechia (0.5), which is in line with the observed difference in genome size. This fact might suggest that *B. ×oycoviensis* originated from *B. pendula* which was 'slightly different' from *B. pendula sensu stricto*, but as already said, further research is needed to draw solid conclusions. Nevertheless, it can be stated that *B. pendula*, *B. ×oycoviensis* and *B. szaferi* in our dataset do not differ genetically at the species level. Ashburner & McAllister [2] argued that the Ojców birch is a weak growing, precocious and heavily fruiting form of *B. pendula*. More recently, the chapter on birches in the Flora of the Czech Republic [21] classified *B. ×oycoviensis* as a variety of *B. pendula*–*B. pendula* var. *oycoviensis*. The experimental data summarized in our study support these opinions.

In Poland, *B. ×oycoviensis* is a species strictly protected by law (Ministry Decree: Poz. 1409 Rozporządzenie ministra środowiska z dnia z dnia 9 października 2014 r. w sprawie ochrony gatunkowej roślin). In Czechia, this taxon is included on the Red List of Vascular Plants, although it is not protected by law [68]. On the other hand, the Ojców birch is included neither on the European Red List of Trees [69] nor on the Red List of Betulaceae [70]. The exclusion of

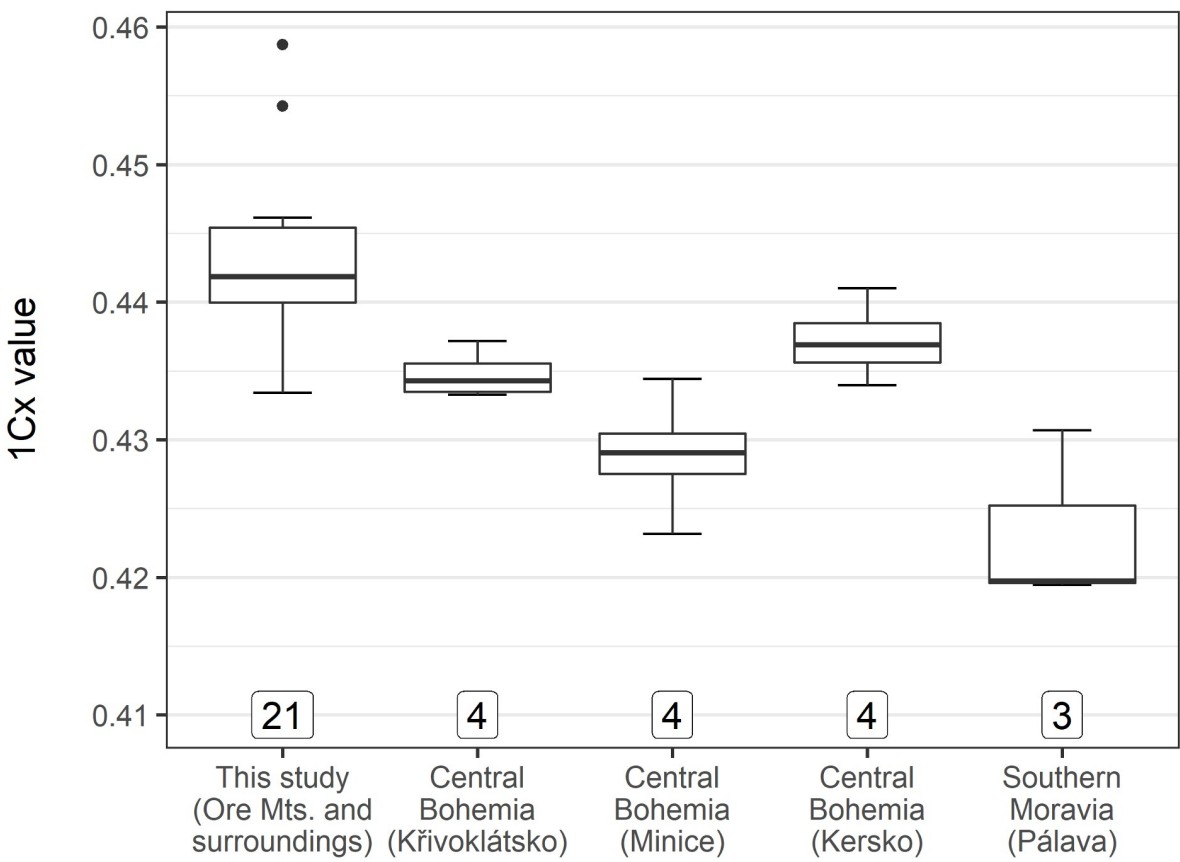

**Fig 7. Comparison of genome size of selected populations of *B. pendula* from Czechia, including some hitherto unpublished data obtained by an identical procedure as described in the material and methods section.** Relatively close locations within Central Bohemia were selected to illustrate the relatively large genome size variability on a small area; samples from Southern Moravia are included as an outlier population. The number of samples analyzed is indicated below each box. The box and whiskers plots are presented in standard Tukey's design: Whiskers depict the minimum and maximum excluding outliers, black dots represent outliers (less than lower quartile—1.5 times the inter-quartile range and more than upper quartile + 1.5 times the inter-quartile range, respectively).

the Ojców birch from the latter two Red Lists might be attributable to its expected downgrading from the species level to a significantly lower taxonomic rank.

Nonetheless, we are convinced that the Ojców birch still deserves some level of protection and distinction in terms of conservation regardless of the taxonomic rank at which this birch is most appropriately classified. It is an interesting birch with a characteristic 'broomy' habitus whose origin is yet to be completely resolved. To answer the remaining questions or to test standing hypotheses related to the Ojców birch and its putative parent (*B. szaferi*), we should keep protecting this rare taxon and its habitats. At present, conservationists are reducing the emphasis on species conservation and are becoming increasingly aware of biodiversity at all levels of the hierarchy of life [64]. We suggest that this should be reflected on some reasonable level also in case of the Ojców birch, at least before its taxonomic position is resolved satisfactorily. In our opinion, this holds true, especially if the rare populations of the Ojców birch in Czechia and Poland diminish in size. A good example of the usefulness of this approach is that of the curly birch (*Betula pendula* Roth. var. *carelica* [Merklin] Hämet-Ahti), whose unique traits are highly valued even though its taxonomic value may be low.

## Conclusions

*Betula pendula* and *B. ×oycoviensis* in our dataset do not differ genetically at the species level despite being distinct morphologically. On a scatterplot produced by the DAPC method, excluding diploid benchmark species (*B. nana* and *B. humilis*), the holotype of *B. szaferi* rests in a cloud consisting of individuals of *B. pendula* and *B. ×oycoviensis*, although it is shifted to the part of the cloud represented by specimens of Czech origin.

Our molecular analyses detected low variability between the groups under comparison (taxa and working units of the Czech and Polish provenance) after the exclusion of the benchmark species. This low variability suggests that the Polish and Czech populations of *B. ×oycoviensis* are genetically very close even though some small differences related to geographic origin may be traced. The classification of the Ojców birch as *B. pendula* var. *oycoviensis* seems more accurate than its treatment as a separate hybridogenous species under the name *B. ×oycoviensis*.

## Supporting information

**S1 Fig. Holotypus of *B. szaferi* sampled in herbarium of the W. Szafer Institute of Botany, Polish Academy of Science (PAS) in Kraków (KRAM 303846).**
(JPG)

**S1 Table. Basic description of all taxa involved in the study.**
(DOCX)

**S2 Table. Estimation of K (the number of 'groups' in microsatellite data) using the method of Evanno *et al*. (2005).**
(XLSX)

**S3 Table. The description of used microsatellite markers.**
(XLSX)

**S1 Data.**
(XLSX)

## Acknowledgments

We would like to thank Jarosław Paluch (University of Agriculture in Kraków, Forestry Faculty) for organizational support and help with the administration of the application of obtaining permits for the sampling and transport of the plant material in and from Poland, Józef Mitka (Botanical Garden of the Jagiellonian University in Kraków) for organizational support and the provision of samples from trees in Botanic Garden, Petr Karlík (Czech University of Life Sciences in Prague) for processing the data from cytometric analyses, Josef Gallo and Josef Petrásek (Czech University of Life Sciences in Prague) for help with sampling in the field, Jakub Baran (Ojców National Park) for organization support, help in the field and sampling in Poland, Jan Rothanzl (Regional Office of Ústí Region, Department of Nature Conservation and Agriculture, Czech Republic) for help with the administration of the application to sample individuals of selested birches in the Ústí Region, Věra Fryčová (ZOO Park Chomutov) for the approval to sample the individuals of *B. ×oycoviensis* in Chomutov, Tomáš Urfus, Jan Bílý and Tomáš Fér (Charles University in Prague) for consultations, and Frederick Rooks for the language editing of our manuscript.

We would also like to thank the Regional Directorate for Environmental Protection in Kraków and the General Directorate for Environmental Protection in Warsaw for granting

permits to sample and transport plant material of birches, Charles University in Prague for enabling our team to conduct laboratory analyses, the W. Szafer Institute of Botany in Kraków, Polish Academy of Sciences for the provision of a leaf fragment of *B. szaferi*, and the Botanical Garden of the Jagiellonian University in Kraków for the provision of birch samples.

## Author Contributions

**Conceptualization:** Ivan Kuneš, Martin Baláš.

**Data curation:** Rostislav Linda.

**Formal analysis:** Rostislav Linda.

**Funding acquisition:** Ivan Kuneš, Martin Baláš.

**Methodology:** Rostislav Linda, Ivan Kuneš, Martin Baláš.

**Project administration:** Ivan Kuneš.

**Software:** Rostislav Linda.

**Supervision:** Ivan Kuneš.

**Visualization:** Rostislav Linda.

**Writing – original draft:** Rostislav Linda, Ivan Kuneš.

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
