## [Decision Letter · Decision Letter 0]

18 Sep 2020

PONE-D-20-26980

Genetic and morphometric variability between populations of *Betula ×oycoviensis* from Poland and Czechia: a revised view of the taxonomic treatment of the Ojców birch

PLOS ONE

Dear Dr. Linda,

Thank you for submitting your manuscript to PLOS ONE. After careful consideration, we feel that it has merit but does not fully meet PLOS ONE’s publication criteria as it currently stands. Therefore, we invite you to submit a revised version of the manuscript that addresses the points raised during the review process.

The main shortage of the manuscript is that the authors used an unsubstantial sample set of the studied taxon. The Material and Methods section lacks clear elaboration of the methods used, in the first place statistical assessment. Inadequately analyzed and presented results may hide many valuable conclusions. It is also recommended to perform additional analysis of the morphometric data (Reviewer #1).

Both Introduction and Discussion sections must be thoroughly revised according to the reviewers' reports. Headings and subheadings should be defined according to the place in the manuscript and the content of the chapter (e.g. three subheadings are stated as: "Morphological analyzes"). 

We look forward to receiving your revised manuscript.

Kind regards,

Branislav T. Šiler, Ph.D.

Academic Editor

PLOS ONE

Journal Requirements:

Reviewers' comments:

Reviewer's Responses to Questions

**Comments to the Author**

1. Is the manuscript technically sound, and do the data support the conclusions?

Reviewer #1: Partly

Reviewer #2: No

2. Has the statistical analysis been performed appropriately and rigorously? 

Reviewer #1: Yes

Reviewer #2: I Don't Know

3. Have the authors made all data underlying the findings in their manuscript fully available?

Reviewer #1: Yes

Reviewer #2: Yes

4. Is the manuscript presented in an intelligible fashion and written in standard English?

Reviewer #1: Yes

Reviewer #2: No

5. Review Comments to the Author

Reviewer #1: The manuscript deals with the endemic taxon of the genus Betula occurring primarily in southern Poland but also in the Bohemian Massif of the Czech Republic. Positive feature of the manuscript is that the authors used several techniques (morphometrics of leaves, flow cytometry and molecular analyses) to describe the taxonomic position of Betula oycoviensis.

As I understood correctly, the Ojców birch is considered to be a putative hybrid species between Betula pendula and B. szaferi and as such it is an endemite for southern Poland. I wonder, how is it possible that this species also occurs in remote parts of Europe in Denmark and Sweden and/or Romania and Ukraine. If the hybrid origin should be true, then in all the above mentioned countries both putative parent species should be present (or they were present in the past). The other option could be that the origin of Ojców birch fits completely into the variation of Betula pendula.

The authors used rather limited sample sizes both for Betula oycoviensis, and for the species they have compared it with, e.g. Betula pendula. I understand that the natural populations of Betula oycoviensis are not very numerous, but on the other hand, they could have done more attempts to produce reliable number of progenies in order to find putative parents. Progeny test based on three individuals is not very trustworthy. The same is true for the individuals growing in the botanical gardens. If you interpret the data based on this experiment, you should be very careful.

As for the morphometrics of the leaves, the information whether the leaves have been chosen from brachyblasts or macroblasts in order to make proper comparison with individuals from the progeny tests is missing. There should not be a problem to get reliable experimental material from high trees (lines 151–153). There are still possibilities to shoot down branches, if there is a problem with climbing the trees. As for the morphometric analyses of leaves, the authors measured a set of linear and angular traits and numbers, thus producing a data set suitable for multidimensional data analyses. However, only PCA is not enough for the morphometric analysis. I would recomend to study the approach published in Koutecký (2015), where the morphometric data were analysed using "MorphoTools" R scripts in R software. Hence, Canonical Discriminant Analyses (CDA) would be employed to select morphological characters which preferably separate ahead selected groups. Also, the nonparametric k-nearest neighbour approach could be used for classificatory discriminant analysis. See: Koutecký P. (2015): MorphoTools: a set of R functions for morphometric analysis. Plant Systematics and Evolution 301: 1115–1121.

More details about the genome size estimation should be given. How many replicates of each samples were carried out? Did the genome size estimation followed “best practice” three replicates on different days as suggested by Doležel et al. (2007) Nat Protoc? The information on the concentrations of mercaptoethanol for Otto II (lines 161–164) is missing. There is no information where the chromosome numbers by which you divided the holoploid (1C) in order to get Cx (monoploid) genome were taken from. Did you count chromosomes? Do you have any plant with known chromosome number to which others measured by flow cytometry were standardized?

I wonder why the author did not use (or at least test) some of the uniparentaly inherited molecular markers (cpDNA) in order to look properly at the origin of putative parents or the way of putative hybridization history.

Major improvement of this manuscript is needed in statistical processing of multidimensional morphometric data e.g. using the discriminant analysis.

Notes

Ln 18 Ojców

Ln 35–37 refrase the sentence

Ln 64 insert comma prior “Czerwona Góra”, i.e. “, Czerwona Góra”

Ln 112 replace “the herbarium of the Polish Academy of Sciences in Kraków” by “herbarium in Kraków” or “Kraków herbarium”. The complete name of this herbarium is already given in lines 100–101

Ln 140–141, 148–149 parts of both sentences are repeated

Ln 162, 171, 184–193 insert space after number (e.g. 20 °C)

Ln 170 Molecular analyses (SSR genotyping)

Ln 172, 174 and others Retsch MM400 should follow the producing company, city and country e.g. DNAeasy Blood and Tissue kit (Qiagen, Hombrechtikon, Switzerland).

Ln 177 microsatellite analysis of nuclear DNA

Ln 199 and 249 Morphometric analysis

Ln 213–214 Information in the first sentence is already given earlier, ln 166–168.

Ln 372, 378 characters or traits instead of parameters

Ln 371–376 You write about the results of univariate tests. What about using the multivariate analysis, e.g. canonical discriminant analysis with subsequent classification of individuals into groups (species and/or population and species).

Ln 473 B. szaferi

Ln 490–492 remove capital letters. Morphological variation among Betula nana (diploid), B. pubescens (tetraploid) and their triploid hybrids in Iceland.

Ln 572 remove “Nonparametric statistics for the behavioral sciences, 2nd ed.”

Ln 573 replace “Mcgraw-Hill Book Company” by “McGraw-Hill Book Company”

Ln 574 remove capital letters. Genetical structure of populations

References

- Complete “doi” where missing, e.g. Järvinen P, et al., 2004; Ln 484 and some other papers.

- Insert a space after year and semicolon

- Insert a space after the volume (or remove it), it should be unified

- Latin names should be in Italics also in References (although this rule is not followed in all papers published in PLoS ONE).

Reviewer #2: Section Introduction:

I have found that the entire "introduction" section is hard to understand clearly. It is not unequivocally clear which taxa are in question. Also, the literature about the locations of taxa of interest is not clearly stated. The introduction section does not clearly describes the problem related to taxonomic status of the Oycow birtch.

Also, in the text of introduction, there is nothing stated about the morphometric and genetic analyses that are used in the research; or the approaches used in research either.

The entire section of the introduction must be revised and supplemented with relevant topics covered in the manuscript.

Subsection Sampling:

It is required to change subsection name into “Plant material”.

Line 85: Is the natural origin of the samples obtained from the botanical garden and zoo known?

Line 108: … although the taxonomic significance of these dark-barked birches remains questioned… This statement does not belong to the section M&M. Move to discussion.

Line 117-119: I suggest to the author consider using the term “population”. Terms “working units” and “group” is confusing. If the author still wants to use these terms, he should better explain and define them and not just refer to another reference (line 105-106).

Subsection Morphological analyses:

Line: 131: …. two branches from each sampled individual were collected for analysis …

the sentence does not agree with the statement in the line 115-116.

Subsection Genome size analysis:

line: 158: This method measures the amount of DNA per cell ( in pg in this case) by comparing the amount of DNA to a known internal standard (in this case Solanum pseudocasicum). Genome size refers to the amount of DNA contained in a haploid genome expressed either in terms of the number of base pairs, kilobases (1 kb = 1000 bp), or megabases (1 Mb = 1 000 000 bp), or as the mass of DNA in picograms (1 pg = 10−12 g)( https://www.sciencedirect.com/topics/immunology-and-microbiology/genome-size ). Please reformulate these sentences with the correct use of the term. This applies to the rest of the text. In general: literature data on the level of ploidy of taxa of interest are mentioned just in the line 214. This information should be stated and commented on in a more transparent way, considering that polyploidization the genus Betula is mentioned.

Subsection Molecular analyses

Line 179: Which 12 microsatellite markers did you use? Are they listed somewhere? You used the microsatellite markers listed in the literature. However, there must be an exact list of which markers you used.

Line 194: You did not determine the "microsatellite length". You determined the length of the amplicons containing the microsatellites. The correct use of the terms is an imperative of scientific work.

Section: Statistical analyses and computations/ subsection Morphological analyses

You did not state how the PCA analysis was performed. How did you use the data given in different units of measurement (mm and degrees of angles) in this analysis? Was the data normalized before PCA analysis? How are they normalized?

How did you analyze continuous and discrete data (eg leaf length and nerve number) together in same PCA analysis?

6. PLOS authors have the option to publish the peer review history of their article (what does this mean?). If published, this will include your full peer review and any attached files.

Reviewer #1: No

Reviewer #2: No

---

## [Author Response · Author response to Decision Letter 0]

16 Nov 2020

Manuscript ID: PONE-D-20-26980

Manuscript Title: Genetic and morphometric variability between populations of Betula ×oycoviensis from Poland and Czechia: a revised view of the taxonomic treatment of the Ojców birch

Dear Editor,

We are grateful to the reviewers for their valuable comments. We appreciate the reviewers’ inputs to the manuscript and the time they spent refereeing it. Their inputs have helped to improve the resubmitted document. We edited the document according to instructions in the review comments to meet the requirements of the reviewers and the publication criteria. The updated version of the manuscript was also completely checked by native speaker specialist in the field (see Acknowledgements).

In the following part, we provide point-by-point responses to the reviewer comments (in italics).

 Authors

Reviewer #1

General comment

The manuscript deals with the endemic taxon of the genus Betula occurring primarily in southern Poland but also in the Bohemian Massif of the Czech Republic. Positive feature of the manuscript is that the authors used several techniques (morphometrics of leaves, flow cytometry and molecular analyses) to describe the taxonomic position of Betula oycoviensis.

As I understood correctly, the Ojców birch is considered to be a putative hybrid species between Betula pendula and B. szaferi and as such it is an endemite for southern Poland. I wonder, how is it possible that this species also occurs in remote parts of Europe in Denmark and Sweden and/or Romania and Ukraine. If the hybrid origin should be true, then in all the above-mentioned countries both putative parent species should be present (or they were present in the past). The other option could be that the origin of Ojców birch fits completely into the variation of Betula pendula.

The authors used rather limited sample sizes both for Betula oycoviensis, and for the species they have compared it with, e.g. Betula pendula. I understand that the natural populations of Betula oycoviensis are not very numerous, but on the other hand, they could have done more attempts to produce reliable number of progenies in order to find putative parents. Progeny test based on three individuals is not very trustworthy. The same is true for the individuals growing in the botanical gardens. If you interpret the data based on this experiment, you should be very careful.

As for the morphometrics of the leaves, the information whether the leaves have been chosen from brachyblasts or macroblasts in order to make proper comparison with individuals from the progeny tests is missing. There should not be a problem to get reliable experimental material from high trees (lines 151–153). There are still possibilities to shoot down branches, if there is a problem with climbing the trees. As for the morphometric analyses of leaves, the authors measured a set of linear and angular traits and numbers, thus producing a data set suitable for multidimensional data analyses. However, only PCA is not enough for the morphometric analysis. I would recomend to study the approach published in Koutecký (2015), where the morphometric data were analysed using "MorphoTools" R scripts in R software. Hence, Canonical Discriminant Analyses (CDA) would be employed to select morphological characters which preferably separate ahead selected groups. Also, the nonparametric k-nearest neighbour approach could be used for classificatory discriminant analysis. See: Koutecký P. (2015): MorphoTools: a set of R functions for morphometric analysis. Plant Systematics and Evolution 301: 1115–1121.

More details about the genome size estimation should be given. How many replicates of each samples were carried out? Did the genome size estimation followed “best practice” three replicates on different days as suggested by Doležel et al. (2007) Nat Protoc? The information on the concentrations of mercaptoethanol for Otto II (lines 161–164) is missing. There is no information where the chromosome numbers by which you divided the holoploid (1C) in order to get Cx (monoploid) genome were taken from. Did you count chromosomes? Do you have any plant with known chromosome number to which others measured by flow cytometry were standardized?

I wonder why the author did not use (or at least test) some of the uniparentaly inherited molecular markers (cpDNA) in order to look properly at the origin of putative parents or the way of putative hybridization history.

Major improvement of this manuscript is needed in statistical processing of multidimensional morphometric data e.g. using the discriminant analysis.

Authors’ answers to general comment

The primary idea that brought us to this topic was the unclarified taxonomic position of Ojców birch that was treated differently by various literature sources. We have used classical morphometric measurements, SSRs genotyping and Genome size analysis. Our aim was to describe the variability of (in particular) the Czech and Polish populations of Ojców birch and compare these two populations together with samples of silver birch. This research has some practical implications for conservation (see the further text). Therefore, we added some short text to the manuscript explaining this.

The topic has two levels, and we intend to address them stepwise (in more studies) although, we admit, these levels are interlinked: The first level is related to the general taxonomical position of Ojców birch. Shortly: Could Ojców birch considered a species, or (much more probably) should this taxon be classified at lower taxonomic rank? Is the Czech population of Ojców birch the same as the population of Ojcow birch in Poland?

The second level may go more in-depth and focus on the hybridization theory and look properly on putative parents etc. Some malformations caused by external factors could theoretically play a role as well, but this could be a topic for another study.

This assessed manuscript aims at the first level and has some practical implications for conservation management of the taxon in the Czech Republic. There is only one locality in Czechia with the acknowledged population of Ojców birch counting several tens of individuals. Some management steps should be taken for its conservation because the Czech population of Ojców birch grows old, and its natural regeneration is poor on the site. However, these steps should reflect the taxonomical position of Ojców birch. To assess this position on level important for nature conservation, the methods used in our study are, in our opinion, sufficient. We suggest that Ojcow birch could not be a species, and this is documented using diploid benchmark taxa in our study (Fig. 5). The applied methods should be sufficient also for the comparison of the Czech and Polish population. Theoretically, Polish individuals could strengthen the Czech population of Ojców birch. For comparison, we have carefully chosen two important locations in Poland, where Ojców birch was reported: Skielek and Kobylany. In Chojnik only four individuals were reported in 1967, Hammernia is a very small nature area (reserve) close to Kobylany, and Czerwena Gora belongs to another district of Poland and sampling there would require a different administrative process to receive a permit for sampling (Ojcow birch is strictly protected in Poland). As for individuals from the botanical garden, we sampled only individuals that manifested the traits of Ojców birch. Lots of trees grown in the botanical gardens are grown from seed, and many of them do not retain the traits of Ojców birch. 

Answer to specific parts of general comment:

…As I understood correctly, the Ojców birch is considered to be a putative hybrid species between Betula pendula and B. szaferi and as such it is an endemite for southern Poland. I wonder, how is it possible that this species also occurs in remote parts of Europe in Denmark and Sweden and/or Romania and Ukraine. If the hybrid origin should be true, then in all the above-mentioned countries both putative parent species should be present (or they were present in the past)...

… Progeny test based on three individuals is not very trustworthy. The same is true for the individuals growing in the botanical gardens. If you interpret the data based on this experiment, you should be very careful.

Betula szaferi is no longer viewed as endemic in Poland (according to Staszkiewicz, 2013); it is, however, considered a parental species of Ojców birch that is allegedly hybridogenous species. Both taxa, B. szaferi and B.×oycoviensis were first described in Poland. Unfortunately, B. szaferi is recently reported as missing in the wild. Even in the Botanical garden in Kraków, there is no longer any living specimen of this taxon, and we did not find any specimen during our detailed survey in Skielek in 2019. Therefore, we used the herbary item of B. szaferi from Kraków (Polish origin) and included the samples of “bred B. szaferi” received from the seed (Czech origin from Volyně) in our research station. We should mention in this regard, that in Kraków, we received a fragment of holotypus (KRAM 303846) for our analyses, see Staszkiewicz (1986) and figure documentation to this answer. The molecular analyses of this holotypus were successful and included in our study. Therefore, we think that our material (despite being limited) provides under the existing circumstances a reasonable base for our manuscript.

In our study, we do not present outcomes of any special progeny test. We included three specimens up to now grown from seeds of the Czech B. oycoviensis in our research station that possess the traits of B. szaferi, next to this we refer to the progeny tests conducted by Jentys-Szaferowa (1967). 

To clarify this and avoid misunderstanding, we changed the term “bred B. szaferi” to “cult. B. szaferi” (as “cultivated”).

We are aware of the limitations and mention these in our manuscript: Ln: 111–114 and 363–368 in the first-round version.

…The other option could be that the origin of Ojców birch fits completely into the variation of Betula pendula…

Yes. Our results suggest this view (e.g. Fig. 5), and we support this opinion in our manuscript: Ln: 424–431 of the original version and the closing sentence of the abstract.

… As for the morphometrics of the leaves, the information whether the leaves have been chosen from brachyblasts or macroblasts in order to make proper comparison with individuals from the progeny tests is missing.

We avoided sampling apical leaves or leaves situated close to the apex of shoots or catkins, we tried to sample well-developed insolated leaves. However, we did not distinguish the position on brachyblast or macroblasts. In the case of B. pendula, to keep sampling from brachyblasts would be difficult, since these are much less frequent (often missing) on the shoots of. B. pendula than in other taxa, at least in case of our samples of. B. pendula. On the other hand, in the case of B. oycoviensis, most of the characteristic small leaves are placed on the brachyblasts. Neither we have recorded any recommendation to distinguish brachyblast or macroblast leaves in the studies focused on distinguishing of birches using the traits of foliar morphology. The word “randomly” was added to the sentence at the beginning of the section “Morphological analyses” to show that we did not distinguish brachyblast vs. macroblast leaves.

There should not be a problem to get reliable experimental material from high trees (lines 151–153). There are still possibilities to shoot down branches, if there is a problem with climbing the trees

Technically, it is the truth. However, according to Polish law, we sampled strictly protected species. To receive permission (which is quite a long-term process), we had to sample a precisely defined amount of material (length of shoots) from each tree, and the technique of sampling had to be described in advance already in the application. We did not dare to mention shooting down the shoots in the application. For tree sampling, we used telescopic shears, which were ca 5 m long, but unfortunately, some trees had insolated part of crowns situated much higher, and we were not able to take samples that were enough representative for foliar morphology. The number of silver birch samples are limited because we have tried to take samples from populations in the close neighborhood of the localities of the Ojców birch, however, at the same time far enough from such localities so that the hybridization could be excluded (or at least were extremely improbable). When implementing the revisions, to make our outcomes in morphology more precise, we measured once more the older samples, so that all the samples were measured by the same person (e.g. Tab. 2, Fig. 2). Relevant statistics were recalculated using updated data.

I wonder why the author did not use (or at least test) some of the uniparentaly inherited molecular markers (cpDNA) in order to look properly at the origin of putative parents or the way of putative hybridization history.

We have used microsatellites for the analysis, as this is widely used approach and also, we already have optimized method (e.g. Kuneš et al., 2019, https://doi.org/10.1371/journal.pone.0224387). 

As for the morphometric analyses of leaves, the authors measured a set of linear and angular traits and numbers, thus producing a data set suitable for multidimensional data analyses. However, only PCA is not enough for the morphometric analysis. I would recomend to study the approach published in Koutecký (2015), where the morphometric data were analysed using "MorphoTools" R scripts in R software. Hence, Canonical Discriminant Analyses (CDA) would be employed to select morphological characters which preferably separate ahead selected groups. Also, the nonparametric k-nearest neighbour approach could be used for classificatory discriminant analysis. See: Koutecký P. (2015): MorphoTools: a set of R functions for morphometric analysis. Plant Systematics and Evolution 301: 1115–1121.

For the morphometric analyses, we have used some parts of MorphoTools script, thank you for the information about this tool. We have provided CDA analysis (partly from the MorphoTools script) as a substitute to previously presented PCA analysis.

…More details about the genome size estimation should be given. How many replicates of each samples were carried out? Did the genome size estimation followed “best practice” three replicates on different days as suggested by Doležel et al. (2007) Nat Protoc? The information on the concentrations of mercaptoethanol for Otto II (lines 161–164) is missing. There is no information where the chromosome numbers by which you divided the holoploid (1C) in order to get Cx (monoploid) genome were taken from. Did you count chromosomes? Do you have any plant with known chromosome number to which others measured by flow cytometry were standardized?

We have done one flow cytometry run per sample. We did not conduct three replicates on different days. There were technical reasons to make only one flow cytometry run per sample: The analyses were not conducted in our facilities but in the laboratory of the Charles University in Prague (see Acknowledgements) to keep the same laboratory and equipment for all analyses in this study. Consequently, there were not as many term opportunities in this laboratory of the Charles University to conduct three replications. Moreover, we analyzed fresh material whose storage time was limited. The sampling and transport of the fresh samples from Poland consumed some significant time. Thus the remaining time available prior to deterioration of samples’ quality did not enable us to conduct analyses on more days. If some doubts about the reliability of the outcomes appeared during processing in the laboratory, the analysis was repeated immediately.

The concentration of mercaptoethanol was added into the text. We did not count chromosomes, we used Solanum pseudocapsicum as an internal standard, as it is stated at the beginning of the “Genome size analysis” subsection of our manuscript. Some of older samples of B. pendula were resampled and analyzed once more using Solanum pseudocapsicum as an internal standard at the beginning of 2020 so that the standard was the same for all samples included in the study. 

Specific comments and answers

Ln 18 Ojców - Corrected

Ln 35–37 refrase the sentence – The sentence was rephrased

Ln 64 insert comma prior “Czerwona Góra”, i.e. “, Czerwona Góra” – Comma inserted

Ln 112 replace “the herbarium of the Polish Academy of Sciences in Kraków” by “herbarium in Kraków” or “Kraków herbarium”. The complete name of this herbarium is already given in lines 100–101 – Changed for “the herbarium in Kraków”

Ln 140–141, 148–149 parts of both sentences are repeated – Thank you for your comment, the second (repeating) sentence was deleted from the manuscript. 

Ln 162, 171, 184–193 insert space after number (e.g. 20 °C) – the space was inserted in all occurrences

Ln 170 Molecular analyses (SSR genotyping) - corrected

Ln 172, 174 and others Retsch MM400 should follow the producing company, city and country e.g. DNAeasy Blood and Tissue kit (Qiagen, Hombrechtikon, Switzerland). – address information added to all described products

Ln 177 microsatellite analysis of nuclear DNA – information added

Ln 199 and 249 Morphometric analysis - corrected

Ln 213–214 Information in the first sentence is already given earlier, ln 166–168. – Sentence deleted

Ln 372, 378 characters or traits instead of parameters – parameters changed for “traits”

Ln 371–376 You write about the results of univariate tests. What about using the multivariate analysis, e.g. canonical discriminant analysis with subsequent classification of individuals into groups (species and/or population and species). – Thank you for your comment, the classification analysis was added, and it helped the study a lot.

Ln 473 B. szaferi - corrected

Ln 490–492 remove capital letters. Morphological variation among Betula nana (diploid), B. pubescens (tetraploid) and their triploid hybrids in Iceland. - corrected

Ln 572 remove “Nonparametric statistics for the behavioral sciences, 2nd ed.” - corrected

Ln 573 replace “Mcgraw-Hill Book Company” by “McGraw-Hill Book Company” - corrected

Ln 574 remove capital letters. Genetical structure of populations - corrected

References

- Complete “doi” where missing, e.g. Järvinen P, et al., 2004; Ln 484 and some other papers. - corrected

- Insert a space after year and semicolon - corrected

- Insert a space after the volume (or remove it), it should be unified - corrected

- Latin names should be in Italics also in References (although this rule is not followed in all papers published in PLoS ONE) - corrected

Reviewer #2

Section Introduction:

I have found that the entire "introduction" section is hard to understand clearly. It is not unequivocally clear which taxa are in question. Also, the literature about the locations of taxa of interest is not clearly stated. The introduction section does not clearly describes the problem related to taxonomic status of the Oyców birch. 

We complemented the introduction section to increase its clarity. We defined the research questions and explained the importance of the answers to these questions.

We added some more information to the map in Figure 1, Table. We equipped the manuscript with an additional table in the supplemental information. 

Also, in the text of introduction, there is nothing stated about the morphometric and genetic analyses that are used in the research; or the approaches used in research either.

The entire section of the introduction must be revised and supplemented with relevant topics covered in the manuscript. 

We have added some more into the Introduction part. However, we think that this may be a topic for the Material and methods section. 

Subsection Sampling:

It is required to change subsection name into “Plant material” 

- corrected

Line 85: Is the natural origin of the samples obtained from the botanical garden and zoo known?

The specimens in the Chomutov’s ZOO originate in Volyně. These were replanted as young trees from the zone of construction of a high-voltage power line. Information was added to the manuscript.

As for the origin of specimens in Krakow, we addressed the local authorities. If we receive the information soon enough, we will add this information to the manuscript.

Line 108: … although the taxonomic significance of these dark-barked birches remains questioned… This statement does not belong to the section M&M. Move to discussion. 

We would prefer leaving the note about taxonomic position here. During the sampling, we have found the dark-barked B. obscura or B. atrata on the surveyed sites. Therefore we decided to include these birches into the analyses, despite the unclarified taxonomic significance of these dark-bared taxa. On the other hand, our study is not aimed at clarification of the taxonomic position of B. obscura or B. atrata. We only want to inform the reader about their inclusion. For this reason, we do not want to open this topic in the discussion section. 

Line 117-119: I suggest to the author consider using the term “population”. Terms “working units” and “group” is confusing. If the author still wants to use these terms, he should better explain and define them and not just refer to another reference (line 105-106). 

Here we used term “working unit” for birches showing mixed traits between B. pendula and B. oycoviensis. In our opinion, the term “population” is not fully accurate for all situations, as some groups of birches technically distinguished in the manuscript in fact are not populations. 

Subsection Morphological analyses:

Line: 131: …. two branches from each sampled individual were collected for analysis…the sentence does not agree with the statement in the line 115-116.

Thank you for your comment, the statement at lines 115 and 116 was corrected; the material for flow cytometry and genetic analyses were both taken from one branch.

Subsection Genome size analysis:

Line: 158: This method measures the amount of DNA per cell (in pg in this case) by comparing the amount of DNA to a known internal standard (in this case Solanum pseudocasicum). Genome size refers to the amount of DNA contained in a haploid genome expressed either in terms of the number of base pairs, kilobases (1 kb = 1000 bp), or megabases (1 Mb = 1 000 000 bp), or as the mass of DNA in picograms (1 pg = 10−12 g)( https://www.sciencedirect.com/topics/immunology-and-microbiology/genome-size ). Please reformulate these sentences with the correct use of the term. This applies to the rest of the text. In general: literature data on the level of ploidy of taxa of interest are mentioned just in the line 214. This information should be stated and commented on in a more transparent way, considering that polyploidization the genus Betula is mentioned.

Thank you for your comment, we have used term “genome size” as in Greilhuber et al. (also cited in the manuscript) and we actually measured the “genome size” in pg (see e.g. Fig 3) and we have used 1Cx values similarly as Greilhuber et al. I think that you mean the same thing by “DNA contained in a haploid genome”, if I get it right.

Subsection Molecular analyses:

Line 179: Which 12 microsatellite markers did you use? Are they listed somewhere? You used the microsatellite markers listed in the literature. However, there must be an exact list of which markers you used. 

We have used the same markers as in cited article. We have added the table with markers as supplementary material.

Line 194: You did not determine the "microsatellite length". You determined the length of the amplicons containing the microsatellites. The correct use of the terms is an imperative of scientific work. 

Thank you for clarification, the term was corrected.

Section: Statistical analyses and computations/ subsection Morphological analyses:

You did not state how the PCA analysis was performed. How did you use the data given in different units of measurement (mm and degrees of angles) in this analysis? Was the data normalized before PCA analysis? How are they normalized?

How did you analyze continuous and discrete data (e.g. leaf length and nerve number) together in same PCA analysis? 

We have removed the PCA analysis and provided new CDA analysis, as it was required by the other reviewer.

Additional requirement

Thank you for including your ethics statement on the online submission form: "Field permit number for sampling and transport of B. humilis:

DZP-WG.6400.6.2020.EP.2 (Generalny dyrektor ochrony Środowiska, Warsaw)

Field permit numbers for sampling and transport of B. oycoviensis:

OP-I.6400.15.2018.KW (Regionalny dyrektor ochrony Środowiska, Krakow) DZP-WG.6400.23.2018.ep (Generalny dyrektor ochrony Środowiska, Warsaw)". 

To help ensure that the wording of your manuscript is suitable for publication, would you please also add this statement at the beginning of the Methods section of your manuscript file.

We have added the statement about field permits in the Material and Methods section.

---

## [Editor Report · Decision Letter 1]

19 Nov 2020

Genetic and morphometric variability between populations of Betula ×oycoviensis from Poland and Czechia: a revised view of the taxonomic treatment of the Ojców birch

PONE-D-20-26980R1

Dear Dr. Linda,

We’re pleased to inform you that your manuscript has been judged scientifically suitable for publication and will be formally accepted for publication once it meets all outstanding technical requirements.

Kind regards,

Branislav T. Šiler, Ph.D.

Academic Editor

PLOS ONE
---

## [Editor Report · Acceptance letter]

23 Nov 2020

PONE-D-20-26980R1 

Genetic and morphometric variability between populations of *Betula ×oycoviensis* from Poland and Czechia: a revised view of the taxonomic treatment of the Ojców birch 

Dear Dr. Linda:

I'm pleased to inform you that your manuscript has been deemed suitable for publication in PLOS ONE. Congratulations! Your manuscript is now with our production department. 

Kind regards, 

on behalf of

Dr. Branislav T. Šiler 

Academic Editor

PLOS ONE